# Life Cycle Environmental Impacts and Health Effects of Protein-Rich Food as Meat Alternatives: A Review

**Maurizio Cellura [1], Maria Anna Cusenza [1], Sonia Longo [1,*], Le Quyen Luu [1] and Thomas Skurk [2,3]**

[1] Department of Energy, University of Palermo, Viale delle Scienze Bld. 9, 90128 Palermo, Italy; maurizio.cellura@unipa.it (M.C.); cusenzam@deim.unipa.it (M.A.C.); lequyen.luu@community.unipa.it (L.Q.L.)

[2] ZIEL Institute for Food and Health, Technical University of Munich, Gregor-Mendel-Straße 2, 85354 Freising, Germany; skurk@tum.de

[3] Else Kröner-Fresenius-Center of Nutritional Medicine, Technical University of Munich, Gregor-Mendel-Straße 2, 85354 Freising, Germany

* Correspondence: sonia.longo@unipa.it

**Abstract:** The food sector is responsible for a considerable impact on the environment in most environmental contexts: the food supply chain causes greenhouse gas emissions, water consumption, reduction in cultivable land, and other environmental impacts. Thus, a change in food supply is required to reduce the environmental impacts caused by the food supply chain and to meet the increasing demand for sufficient and qualitative nutrition. Large herds of livestock are inappropriate to achieve these goals due to the relevant impact of meat supply chain on the environment, e.g., the land used to grow feed for animals is eight times more than that for human nutrition. The search for meat alternatives, especially for the intake of critical nutrients such as protein, is a consequent step. In the above context, this paper summarizes the health aspects of protein-rich food alternatives to meat and carries out a literature review on the life-cycle environmental impacts of this alternative food.

**Keywords:** life cycle assessment; proteins; food; environmental sustainability human health

## 1. Introduction

Proteins are a unique source of amino acids, the building bricks of life. There is constant formation and breakdown of proteins in our body. Therefore, their uptake is essential but requires specification concerning the amount and quality. According to various nutrition associations, a minimum of 0.8 g/kg body weight is recommended for the daily intake in the age group of 19–65 year-old adults. Above 65 years, 1.0 g/kg is recommended [1]. This amount is mainly required to preserve fat-free lean mass. The essential amino acid leucine thereby indicates the biological value in muscular protein synthesis [2]. It is noteworthy that higher protein intake is recommended for athletes but also for reducing body weight or under pathophysiological conditions such as severe/end-stage nephropathy [3–5]. Moreover, high protein intake was investigated concerning the suppression of appetite and food intake to prevent body weight gain in children [6]. Proteins are therefore essential macronutrients, and animal-derived foods are considered the most efficient source, according to their higher content in essential amino acids such as leucine. However, due to the negative impact of storage conditions on the environment, adequate plant-based substitutes have to be considered in the future.

Therefore, a healthy reference diet was suggested to consist basically of plant-based sources, such as vegetables, fruits, whole grains, legumes, and others, and only low amounts of meat [7]. Soybean, peas, quinoa, rice, green beans, faba beans, lentils, and lupine are among the foods that can be used as protein sources for alternatives to meat.

Choosing a diet with an adequate amount of proteins must consider both the nutritional and healthy aspects of foods. Furthermore, another element indirectly linked

with human health and well-being must be considered, that is, the impact of food on the environment. The food supply chain is globally one of the primary users of several natural resources. Food production depends on land, biodiversity, minerals (such as nitrogen and phosphorus), fresh water, and marine resources [8]. All food system activities from farming to harvesting, processing, packaging, distribution, and cooking depend on energy, which currently is mainly derived from fossil fuels. In addition, the food sector generates a large amount of waste, and it is one of the major drivers of several environmental impacts, such as the loss of biodiversity, soil degradation, water depletion, and greenhouse gas (GHG) emissions.

Agricultural land area (including pastures for grazing livestock) accounts for 38% of the global land surface [9]; agriculture is also the significant driver of land-use change [10]. According to the results of a study of the European Commission Joint Research Centre, the food system handles a third of global GHG emissions [11]. Another study attributes a percentage ranging from 21% to 37% of overall anthropogenic GHG emissions to the food system [12]. Irrigated crops sustain 40% of the global food production, accounting for 70% of all water withdrawn from aquifers, streams, and lakes [13]. Moreover, according to Poore and Nemecek (2018), the food system causes about 30% of global terrestrial acidification and 80% of eutrophication (freshwater and marine) [14].

The need for a more sustainable food production system is globally recognized. Due to the expected population growth, the pressure on natural resources from food system activities will increase. In addition, the global food system, becoming more energy-intensive [11], and the dietary shifts towards more resource-intensive products will increase these pressures even more, leading to a greater environmental impact [8].

Food systems sustainability is at the heart of the United Nations 2030 Agenda for Sustainable Development [15]. Specifically, the second Sustainable Development Goal aims at ending hunger and achieving food security and improved nutrition, while ensuring the reduction in environmental impact of food. In the European Union, the Farm to Fork strategy and its broader European Green Deal policy addresses the challenges to reduce the environmental and climate footprint of the European food system [16,17].

Meeting food demand in an environmentally sustainable way needs to rely on integrated methods that allow for identifying the hotspots of the whole food system life cycle, for comparing possible alternatives, and for avoiding burden shifting geographically, temporally, and along supply chains [18]. In this context, the Life Cycle Assessment (LCA) methodology is widely applied by the scientific community to analyze the environmental impacts of the food system's supply chain, with the aim of supporting the identification of sustainable solutions for global food challenges [19,20]. LCA supports the identification of the supply chain hotspots and the comparison of different agricultural practices. It provides a depth insight into the environmental effects associated with a full diet or with a full meal [21] and on dietary shifts, since they may cause the burden to shift from one stage to another in the food supply chains of different foods, or from one impact category to another [22]. LCA also provides insight into food waste prevention [23,24] and food waste handling processes [25–27].

Considering the complexity of the food supply chain, applying the LCA methodology, covering several categories of impacts, and embracing different life cycle stages of production and consumption is paramount for achieving food sustainability goals.

In the above context, the goal of this paper is two-fold:

- to describe the healthy characteristics of protein-rich food that can be used as an alternative to meat (Section 2) by highlighting the advantages of this food as a substitution for meat; and
- to carry out a literature review on the life cycle impacts of protein-rich food alternative to meat and the potential environmental benefits of substituting meat with other food giving the same amount of proteins (Section 3).

## 2. Health Characteristics of Animal vs. Plant-Based, Protein-Rich Foods

Despite the essential nature of many amino acids, the long-term impact of protein intake beyond a certain threshold on health is unknown. As higher protein intake is known to affect the growth hormone receptor/IGF-1 axis, it seems to be associated with a higher susceptibility to tumor diseases, diabetes, and overall mortality, especially in a population younger than 65 years [28]. It is noteworthy that many studies on this tumor-promoting activity relate to protein from animal sources and involve mainly the gastrointestinal tract. Despite some inconclusive results on colon cancer (CC) [29], the International Agency for Research on Cancer (IARC) recently classified processed meat (meat transformed through salting, curing, fermentation, or smoking) as carcinogenic and red meat (beef, veal, pork, lamb, mutton, horse, and goat) as probably carcinogenic [30]. Observational studies indicate an increased risk for colorectal cancer (CRC) and CC with beef and lamb, but not with pork and poultry [29]. It was calculated that each 100 g/day intake of red and processed meat corresponds to a 12% higher risk of colorectal cancer (CRC) [31]. A dose-dependent effect for red and processed meat was also shown for gastric cancer [32]. Cardio metabolic outcomes were investigated and found weak associations for red and processed meat [33]; on the contrary, white meat (e.g., from poultry) seems slightly beneficial [34].

A yet-underestimated risk factor for CC might be the modifying effect of the intestinal microbiota, depending on the diet. However, the studies performed to date cannot separate the causes or consequences of the experimental findings [35]. Shot gut metagenomics determined that the microbiome is associated with increased availability of secondary bile acids as a risk factor for CRC [36]. Taken together, it seems therefore advisable to reduce protein intake from animal meat including processed products such as offal and blood.

In contrast, various available reports have indicated that a plant-based diet seems protective against various cancer diseases, including hepatocellular carcinoma or gastric, colonic, and breast cancer [31,37–39]. Therefore, the EAT-Lance Commission suggested a "universal health reference diet", which consists basically of plant-based sources, such as vegetables, fruits, whole grains, legumes, and others, and only low amounts of meat [7]. In this context, it is important to highlight that the source of protein intake substantially changed between the 1960s and today from predominant plants to animal sources [40]. Even though there is no agreement on how much meat consumption has increased in the last few decades, 350–500 g/week is considered sufficient [41].

Documented health benefits of plant-based diets include, e.g., the cardiovascular system [42], more specifically, dyslipidaemia [43] or ischemic heart diseases. Moreover, a reduced total cancer mortality of about 15% was reported [44]. Beyond cancer, beneficial data on vegetarian or vegan diets are also available concerning the management of metabolic diseases such as type 2 diabetes [45], and even the substitution of red meat with poultry or fish reduces the risk [46]. Specific health benefits were also published for soy products resulting in a lower risk of gout [47], lower CHD risk [48], and a lower risk of prostate and breast cancer [49,50]. These beneficial effects on mortality seem to be dose-dependent [51]. In this context, it is noteworthy that some phytoestrogens, such as from soy, might have opposing health effects, as they are classified as endocrine disruptors [52]. However, recent data show that the impact on male reproductive hormones is low or absent [53]; they may even have tendentially beneficial effects on the risk of prostate cancer [54] and breast cancer mortality and cancer recurrence [55].

The search for meat alternatives has to take into account protein qualities, as amino acid composition affects physiological effects. A higher-quality value is assumed if the amino acid pattern is as similar to the cellular pool of proteins in the body. Eggs were therefore given a value of 100. Therefore, on average, most vegetable proteins have a lower biological equivalence. In order to assess protein quality, first, the amount of essential and branched-chain amino acids has to be considered. In general, animal protein is a more effective source compared to plant proteins, as plant proteins may contain anti-nutritive factors to a varying extent, including trypsin inhibitors, phytates or glucosinolates, and others [56], which might reduce bioavailiabilty. Trypsin inhibitors are present in legumes,

cereals, and soybeans [57]. Therefore, vegetable protein quality is considered low, as amino acids from those proteins are not readily available for, e.g., muscular protein synthesis. This difference is reflected by the digestible indispensable amino acid score (DIAA), which gives higher values for the animal than for plant proteins [56]. However, a combination of various protein sources (P4), including plant proteins (soy and peas), might mitigate the restricted value of pure (soy) protein and could be shown to be similar to casein or whey [58]. In addition, probiotic supplementation could be a valuable nutritional strategy to improve postprandial changes in blood amino acids and to overcome compositional shortcomings of plant proteins [59].

An important functional readout for protein quality is the exercise-induced protein synthesis rate, especially in older adults [60]. Alternatives to meat must therefore be rated accordingly. Quinoa protein seems a valuable choice [61], and soy protein does not seem to be significantly inferior compared to animal protein [62], despite a lower leucine content [63]. Pea protein showed yet unclear results on post-exercise muscle damage and repair [64]. Another alternative to meat involves insect protein, which results in similar postprandial blood amino acid profiles as soy protein but elicits lower blood concentrations than whey. During eight weeks of resistance training, insect protein showed similar effects on muscle mass and strength compared to carbohydrate supplementation [65,66]. Muscle protein synthesis rates after intake of mealworm protein supplementation were similar to a milk protein concentrate [67]. Notably, the allergy-inducing potential of insect proteins has to be considered [68]. Due to its consistency and processability, seitan, a gluten-based product from wheat, also seems a potential meat alternative. Seitan is highly digestible; however, its low content in the essential amino acid lysine results in a low protein quality score [69]. At the same time, due to its origin, it is not suitable for patients with celiac disease. Products with an admixture of seitan are well-accepted [70], despite the fact that taste and texture might be aspects to decline meat alternatives.

Although comparative studies on protein content and different quality scores of meat alternatives are available [63,71–73], for most alternative proteins, physiologic equivalence studies are largely missing, and even technological aspects are rarely addressed [74]. To support the substitution of meat by alternative protein sources in a viable sustainable diet, further interventional studies must address health aspects respecting age-depending needs and acceptance.

## 3. Life Cycle Assessment of High Protein Food as Alternative to Meat

With regard to health and nutrient aspects, animal proteins can be substituted by plant-based, high-protein food alternatives. This section compares and verifies the substitution of high-protein food as an alternative to meat with regard to the environmental aspect. This is undertaken by describing the collection and reviewing papers on LCA of high protein food that can be used to replace meat. A description of the methodology for conducting the review is reported, to clarify how and which papers are selected. After that, the reviewed case studies are summarized, focusing on their methodological aspects and obtained results, and a comparison with the environmental impacts of meat is made.

### 3.1. Methodology Description

The search for literature was performed on Science-direct in March 2021 using the keywords "LCA, high protein food, meat replacer". In order to avoid missing LCA papers specifically focused on different types of legumes or seeds, other keywords including pluses, soybean, and quinoa were searched for. Only review and research articles were selected, excluding encyclopedia and book chapters. The papers' titles and abstracts were skimmed through to exclude papers that (1) were irrelevant to the life cycle approach and (2) studied the following product systems: biofuel, food waste, food service, food preservation, pet food (animal feed), and meat (fish, prawn, dairy, honey, and insect-based) products. At the end of the skimming, there were 28 papers published in the period between 2009 and 2021,

relevant to LCA of high protein food that can be used to replace meat to provide similar nutrients. Figure 1 presents the number of papers by year.

**Figure 1.** Relevant papers by years.

These 28 papers include 4 review articles, 2 methodology articles, and 22 case studies on the environmental impacts of legumes (soybeans, beans, peas) and plant-based meat replacers (quorn, mycoprotein). The case studies' topics are presented in Table 1.

**Table 1.** Topics of 22 reviewed LCA case studies.

| Topics | Number of Case Studies [1] |
|---|:---:|
| Various types of plant-based products to replace meat | 4 |
| Soy/soybean | 6 |
| Bean | 4 |
| Pea/Chickpea | 5 |
| Protein concentrate | 5 |
| Quorn [2] | 1 |
| Quinoa | 1 |
| Lentils | 1 |

[1] The total number of case studies is larger than 22, as there are several papers studying more than one product system. [2] Quorn is a meat replacer, made from cultured fungus, mixed with egg albumin as binder.

It is common that there are at least two product systems being simultaneously studied in one paper (15 out of 22 case studies). These studies either conducted LCAs of various types of high protein products to replace meat or compared life cycle environmental impacts of meat and alternative plant-based products. At the end of the searching and selecting process, 22 LCA case studies on high-protein food to replace meat were reviewed, as they provided adequate information on applied methodology and obtained results on life cycle environmental impacts. Other articles, including review and methodology articles, which do not provide information on the environmental impacts of food, have not been reviewed, but they were mentioned or analyzed in this paper.

*3.2. Papers on Alternative High Protein Food*

This part is the summary of reviewed case studies. For each case study, the research topics, functional units (FUs), system boundaries, allocation procedures, important assumptions, and obtained results are described.

Davis et al. (2010) developed an LCA to analyze the potential environmental benefits of introducing more grain legumes in human nutrition [75]. Four meals served in Sweden

and Spain, with different amounts of soybeans or peas, were analyzed, including soy pork chop, pea pork chop, pea-meat sausage, and pea burger. For soy pork chop and pea pork chop, soy and pea are used as feed for swine. Pea in combination with other meat-based ingredients is used to made pea-meat sausage; meanwhile, the pea burger is made from exclusively pea. The functional unit is a meal served in households. The protein contents of the four meals are slightly different, ranging from 33.7 g to 34.8 g per meal. The study covered the cradle-to-grave system boundaries, from raw material extraction to the end of life, including packaging and waste management of used packaging. Five environmental impacts were assessed, including global warming potential (GWP), eutrophication potential (EP), acidification potential (AP), land use, and energy consumption. It was identified that pea burgers were more environmentally favorable than meat [75].

Specifically, GWP per one meal of pea burger served in Sweden and Spain (from 0.54 to 1.16 kg $CO_{2eq}$) is lower than that of the other foods of about 33–56%, while EP is about half (from $5.7 \times 10^{-3}$ to $1.04 \times 10^{-2}$ kg $N_{eq}$). AP per one meal of soy pork chop (from $1.11 \times 10^{-2}$ to $2.18 \times 10^{-2}$ kg $SO_{2eq}$), pea pork chop (from $1.08 \times 10^{-2}$ to $2.15 \times 10^{-2}$ kg $SO_{2eq}$), and pea-meat sausage (from $1.17 \times 10^{-2}$ to $2.16 \times 10^{-2}$ kg $SO_{2eq}$) is also higher than that of pea burger (from $2.38 \times 10^{-3}$ to $9.98 \times 10^{-3}$ kg $SO_{2eq}$). One pea burger meal required from 16.1 to 20.2 MJ of energy, which was similar to the amount of energy for the other meals (about 18–24 MJ). The authors reported that land use for pea burger was lower than that of the other meals, but numerical results were not included in the study [75].

Knudsen et al. (2010) assessed the environmental impacts of imported organic soybeans from China to Denmark, taking into account the production of agricultural inputs, the cultivation process, and the sorting, packaging, and transport to fodder companies in Denmark for one tonne of soybean [76]. The results indicated that 51% of GHG emissions came from transportation stages, 35% from agricultural activity, and the remaining from inputs of production and processing stages. Organic soybeans showed better performance than conventional ones in terms of GWP (156 kg $CO_{2eq}$ vs. 263 kg $CO_{2eq}$), non-renewable energy use (773 MJ vs. 1710 MJ), AP (2.3 kg $SO_{2eq}$ vs. 4.5 kg $SO_{2eq}$), and EP (5 kg $NO_{3eq}$ vs. 13 kg $NO_{3eq}$), while worse values are observed in terms of land use (0.36 ha vs.0.32 ha) [76].

Zhu and Ierland (2004) compared the production and consumption chains of pork and alternative protein food from dry pea using LCA [77]. Five impact indicators were assessed per one tonne of protein content for both product systems (GWP, AP, EP, water consumption, and land use). One tonne of protein content was equivalent to 5.5 tonnes of pork and to 10 tonnes of dry peas. It was identified that pork emitted more and consumes more resources than the alternative protein food in all the examined impact categories. Specifically, pork chain contributed to 675 kg $NH_{3eq}$, which was 61 times higher than the impact from the alternative protein food. Similarly, GWP and EP of pork chain were 6.4 times (77,883 vs. 12,260 kg $CO_{2eq}$ per FU) and 6 times (2491 vs. 417.3 kg $N_{eq}$ per FU) higher than those of dry peas, respectively. Pork required 3.3 times more fertilizers and 1.6 times more pesticides than the pea protein food. It consumed 36,152 $m^3$ of water and occupied 5.5 ha of land, which were 3.3 times more water and 2.8 times more land than the pea protein food [77].

Smetana et al. (2015) compared the environmental performance of different meat substitutes (dairy-based, lab-grown, insect-based, gluten-based, soy meal-based, and mycoprotein-based products) with chicken [78]. The system boundaries were from cradle to plate, excluding recycling of packaging and waste treatment after human consumption. It was assumed that transportation distance to the consumer was 10 km, and food was fried on an electrical stove. The results per one kilogram of ready-to-eat meal at consumer indicated that lab-grown meat and mycoprotein-based products had the highest impacts, followed by chicken and dairy-based, and gluten-based meat substitutes. Insect-based and soy meal-based substitutes had the lowest impacts [78].

Specifically, the climate change impact for one kilogram of chicken (5.2–5.82 kg $CO_{2eq}$) doubled that of soy meal-based products (2.65–2.78 kg $CO_{2eq}$) but was comparable to that of mycoprotein-based products (5.55–6.15 kg $CO_{2eq}$). Similarly, the non-renewable energy

consumption for chicken was 51.64–63.4 MJ, lower than that of mycoprotein-based products (60.07–76.8 MJ), but much higher than that of soy meal-based products (27.78–36.9 MJ). Meanwhile, land use of chicken (3.85–3.89 m$^2$) was much higher than that of soy meal-based products (1.06–1.44 m$^2$) and mycoprotein-based products (0.79–0.84 m$^2$). For the alternative FU of 3.75 MJ energy content with corrected weight, lab-grown and mycoprotein-based products had the worst performance, whereas insect-based, soy meal-based substitutes, and chicken had the best performance; other substitutes had medium impacts. Regardless of FUs, soy meal-based substitute had the lowest impact. Lab-grown meat had the highest impacts in most categories, except for agricultural land occupation [78].

　　Sturtewagen et al. (2016) assessed the nutrition value and resource consumption of six specific meals of pork and quorn produced in different countries of Belgium and United Kingdom [79]. Resource consumption was measured by the cumulative exergy extraction from the natural environment (MJ$_{ex}$). The meals of pork cooked at home without recycling frying oil had the highest resource consumption, while quorn cooked at a canteen had the lowest resource consumption. For all meals, biomass resources contribute 52–64% of the total resource consumption, mainly due to land occupation. The resource consumption results per kilogram of pork and quorn were largely different, at about 149 and 85 MJ$_{ex}$, respectively. When the protein and energy amount were taken into account, the nutritional content was highest for quorn cooked at home without recycling frying oil, while pork cooked at home without recycling frying oil had the lowest nutritional content. Consequently, this changed the relative difference in the resource consumption results per FUs based on the protein and energy amount. Per 100 g of protein and per 100 kcal, the differences in the results were not significant: about 70 and 63 MJ$_{ex}$ per 100 g of protein, and about 13.2 and 10.6 MJ$_{ex}$ per 100 kcal from pork and quorn, respectively [79].

　　Lathuillière et al. (2017) conducted a LCA of soybean in Brazil for calculating the land use impacts [80]. The production systems of soybean and maize was examined, but all impacts were allocated to soybean. In the study, the authors excluded important changes in the landscape, such as consideration of pasture as natural vegetation and indirect land use change. It was concluded that, depending on the yield, soil type, biomass above the ground, and regeneration times between biomes, the impacts of soybean in different regions of Brazil were different. Specifically, land transformation and land occupation impacts on the biodiversity were $1.17 \times 10^4$ m$^2$y and $2.93 \times 10^3$ m$^2$y, respectively, per tonne of soybean produced in the Amazon, and $8.04 \times 10^3$ m$^2$y and $2.75 \times 10^3$ m$^2$y for Cerrado. The land use impacts were converted into economical costs that must be paid so that the land would fulfil the same ecosystem service as it was before any change was made to its ecological functions. The total land transformation and land occupation damage in Amazon would cost USD 532 and USD 260, which were much larger than those in Cerrado, at USD 231 and USD 153 per tonne of soybean. The hotspots of the analysis were the mechanical filtration properties of soil, the land climate regulation, and the biotic production potential [80].

　　Mierlo et al. (2017) applied the life cycle approach to find out the composition of meat replacers to chicken and beef, with equivalent nutrition, while minimizing their impacts on GWP, water use, land use, and fossil fuel depletion [81]. Four types of meat replacers were studied, including vegetarian (containing plant-based ingredients and some animal-based ingredients, without any insect), vegan (containing only plant-based ingredients), insect-based (containing only insect-based ingredients), and fortification-free (containing plant-based ingredients, without any supplement of minerals and vitamins). Two system boundaries were considered. The first one covered from the agricultural stage to the processing-to-ingredients stage (ingredient system boundaries), while the second one also included the processing stage from ingredients to end-products (end-product system boundaries).

　　When considering the product system within the ingredient system boundaries, soy was the preferred ingredient to fulfil nutrient supply as well as to minimize environmental impacts. It was included in all four types of meat replacers at different percentages. For example, for the vegetarian/vegan chicken replacers, the smallest total environmental

impacts were obtained with a meal containing 1.7% of soy flour, 42.2% of soy protein concentrate, 3.8% of soy protein isolate, 52.2% of water, and less than 0.1% of B12 fortification. In this case, GWP was 0.62 kg $CO_{2eq}$, land use was 2.64 $m^2$/year, water use was 0.04 $m^3$, and fossil fuel depletion was 9.32 MJ per kg of vegetarian/vegan chicken replacer. When the system boundaries were extended to the end-products, the vegan meat replacers had the largest potential for impact reduction for all the indicators except for water use. The insect-based meat replacer was the most promising option to reduce water consumption but showed relatively high fossil fuel depletion values. Regarding GWP, the end-product processing stage contributed about 26% (for chicken replacer) and 30% (for beef replacer) to the total impact. Meanwhile, this stage consumed a relatively low amount of the water (14% of total water use for chicken replacer and 16% for beef replacer) [81].

Cancino-Espinoza et al. (2018) analyzed the environmental impacts of organic quinoa supply chain in Peru for export [82]. The system boundaries covered farming activities (soil tillage, sowing, harvesting, and applying fertilizers and pesticides), post-harvesting activities (drying, cleaning, classifying, and packaging), and distribution of final product. GHG emissions, mainly caused by on-field emissions due to fertilization (58.5%), were 441 g $CO_{2eq}$ per 500 g package of organic quinoa, ready for sale. Compared to other high-protein foods, especially of animal origins, the environmental impacts per gram of protein were considerably low, at 6.47 g $CO_{2eq}$ vs. 34.10 g $CO_{2eq}$ for rice, 134.88 g $CO_{2eq}$ for beef, 40.63 g $CO_{2eq}$ for pork, and 19.25 g $CO_{2eq}$ for chicken [82].

Lee and Choe (2019) studied the energy performance and GHG emissions of organic and conventional small-scaled soybean farms in South Korea, following a "from cradle to farm gate" approach [83]. The results highlighted that conventional farming was significantly more energy efficient than organic farming (13,416 MJ/tonne of soybean vs. 22,421 MJ/tonne). GHG emissions of the organic soybean farming systems were about 20% higher than that of the conventional system (2045.11 kg $CO_{2eq}$/tonne vs. 1657.35 kg $CO_{2eq}$/tonne). The hotspots were manure, fertilizers, and fuel, which accounted for more than 85% of the total GHG emissions [83].

Ilari et al. (2019) assessed the impacts of one kilogram of frozen green bean, from the crop production transportation and industrial processing [84]. The FU caused an impact on Abiotic Depletion Potential (ADP) and ADP fossil of $2.2 \times 10^{-6}$ kg $Sb_{eq}$ and 9.5 $MJ_{eq}$, respectively. The results for the environmental impacts are as follows: 0.74 kg $CO_{2eq}$ for GWP; $8.1 \times 10^{-8}$ kg $CFC_{11eq}$ for ozone layer depletion potential (ODP); 0.13 kg $1,4DB_{eq}$ for human toxicity potential (HTP); 0.12 kg $1,4DB_{eq}$ for freshwater ecotoxicity potential (FETP); $1.9 \times 10^{-4}$ kg $C_2H_{4eq}$ for photochemical oxidation potential (PCOP); $3.5 \times 10^{-3}$ kg $SO_{2eq}$ for AP; and $1.7 \times 10^{-3}$ kg $PO_4{}^3{}_{eq}$ for EP. The energy use in the industrial processing phase largely contributed to ADP fossil, GWP, and ODP, while the crop production phase is the main contributor to ADP, HTP, FETP, AP, and EP [84].

Based on three balanced dietary patterns of omnivorous, vegetarian, and vegan, Corrado et al. (2019) assessed how personal consumption choices and behavior can affect GHG emissions [85]. The system boundaries covered the stages of food production, cooking, and food waste. A large part of GHG emissions came from the supply chain stages prior to consumption, up to 66–71% of the total. The cooking phase and food waste treatment accounted for 15–21% and 11–13%, respectively. Specifically, to provide energy for one person per day, equivalent to 2131 kcal, omnivorous diet emitted 3.24–3.92 kg $CO_{2eq}$ (mean ± standard deviation). The emissions of vegetarian and vegan diets were lower, 2.76–3.2 kg $CO_{2eq}$ and 2.61–3.13 kg $CO_{2eq}$, respectively [85].

Heusala et al. (2020a) assessed the potential environmental benefits of adding oat protein concentrate (OPC) to regular food products, including wheat bread (made from wheat flour), durum pasta (made from durum wheat semolina), and yogurt [86]. For the FU of one kilogram of product, adding OPC to regular food products generally increased the carbon footprints (CFs) and land use impacts. Specifically, the CF of wheat flour was 0.5 kg $CO_{2eq}$, which was half as much as that of durum wheat semolina and oat starch (both at 1 kg $CO_{2eq}$), and a sixth of OPC (at 3.3 kg $CO_{2eq}$). Land use of wheat flour and

durum wheat semolina was 2 and 4.4 m$^2$ respectively, while the impact of oat starch and OPC was 1 and 3.2 m$^2$, respectively. For the FU of one kilogram of protein, adding OPC to regular food slightly increased the CF from 4.8 kg CO$_{2eq}$ for wheat flour and 7.8 kg CO$_{2eq}$ for durum wheat semolina to 8.8 kg CO$_{2eq}$ for OPC (except for the CF of oat starch pasta, which is 12.4 kg CO$_{2eq}$). Land use impacts of all products decreased from 18.2 m$^2$ of land for wheat flour and 33.5 m$^2$ for durum wheat semolina to 12.1 m$^2$ for oat starch and 8.6 m$^2$ for OPC [86].

CFs per kg of protein of all high-in-protein versions of food containing OPC were as similar as those of common versions of food such as pasta, bread, and yogurt. For example, the CF of pasta made with oat starch at 20% of protein was 12.1 kg CO$_{2eq}$, which was similar to that of pasta made with durum wheat semolina at 5% of protein (12.8 kg CO$_{2eq}$), with durum wheat semolina and OPC at 12% of protein (at 12.2 kg CO$_{2eq}$), or with durum wheat semolina and OPC at 20% of protein (at 11.5 kg CO$_{2eq}$). Furthermore, the authors found that food products containing OPC had a smaller CF per kg of protein when compared to animal-based protein sources and a similar CF when compared to naturally protein-rich plant products such as legumes. The production of ingredients used in the food (durum wheat semolina, wheat flour, and OPC) contributed to 60–90% of the impact, while energy consumed in the food production and cooking contributed to 10–40% [86].

Heusala et al. (2020b) also conducted a LCA to compare the environmental impacts of OPC with faba bean protein concentrate (FBC) and other food products [87]. In this study, the authors applied the same system boundaries of cradle-to-processing stage as the previous one. Per kg of product, CF of OPC (3.3 kg CO$_{2eq}$) was higher than that of FBC at both low and high yields (around 1.1–2 kg CO$_{2eq}$). In contrast, land use of OPC was much smaller than that of FBC (3.2 and 8–20.8 m$^2$ per kg of product, respectively). Additionally, per kg of protein, the CFs of both OPC (8.8 kg CO$_{2eq}$) and FBC (1.9–3.4 kg CO$_{2eq}$) were lower (50% and 80–90%, respectively) than that of dairy protein. OPC had four times higher CF than legume protein sources. Land use of OPC was much smaller than that of FBC, at 8.6 vs. 13–35m$^2$ per kg of protein. Different from OPC, the hotspot of FBC was the cultivation of faba bean, which contributes to 55–76% of CF [87].

Mogensen et al. (2020) compared the GHG emissions and land use impact of standard diet with diets containing OPC for the whole food chain, from primary production to food intake [88]. The study took into account the energy used for cooking at home, as well as unavoidable food losses and weight changes after cooking. The impacts were quantified per one FU of daily intake of food and beverages for one adult, being equivalent to 8831–9986 KJ per day. The results indicated that the CF of standard diet was 4.07 kg CO$_{2eq}$, which would reduce to 3.95 kg CO$_{2eq}$ if OPC bread, pasta, and oatgurt (yogurt made from oat instead of milk) were used, while the land use remained the same at 5.69 m$^2$. It should be noted that the OPC-based diet provided higher protein than the standard diet, as OPC bread, pasta, and oatgurt had the higher protein content per weight compared to conventional wheat bread, durum pasta, and yogurt [88].

Additionally, the authors developed several scenarios to compare standard diets with OPC-based diets at the same amount of protein. In a scenario called animal-based protein replacement, OPC bread, pasta, and oatgurt were used as substitutes for wheat bread, durum pasta, and yogurt, and all animal-based food was reduced so that this diet provided the same protein amount as the standard diet. GHG emissions from this animal-based protein replacement diet reduced by 8%, and land use decreased by 14% compared to the standard diet. In another scenario called beef and pork replacement, the OPC-enriched food items were used in bread, pasta, and oatgurt, and the amounts of beef and pork were reduced provided that the same protein amount as the standard diet was obtained. GHG emission and land use from the beef and pork replacement diet reduced by 13% and 26%, respectively, compared to standard diet [88].

Escobar et al. (2020) quantified the GHG emissions related to the production (from cradle to processing) and trade of exported Brazilian soy [89]. The average GHG emissions were 0.69 $tCO_{2eq}$ per tonne of Brazilian soy. A comparison among soy-growing municipalities indicated that GHG emissions of soy from different municipalities was significantly variable, ranging from 0.13 to 29.47 $tCO_{2eq}$ per tonne of soy. The soy from some specific municipalities within Pará and Matopiba States had higher GHG emission intensity due to land transformation from forest to agricultural land. In these municipalities, land use change contributed from 72% to 87% of the total GHG emissions [89].

At an international scale, the largest GHG emissions intensity were related to soy exported to EU, at 0.77 $tCO_2eq$ per tonne of soy. The emission intensity of soy exported to EU is significant, as the soy was grown in municipalities with many deforestation activities, e.g., land use change from forest to land for growing soy. The GHG emissions intensity of soy exported to China, the largest soy importer, were 0.67 $tCO_{2eq}$ per tonne of soy. The emission intensity of soy exported to China is lower than that of soy exported to the EU, as most of China's imported soy came from municipalities with average GHG emission intensity such as Mato Grosso. Total GHG emissions from Brazilian soy exports in 2010–2015 were estimated at 223.46 $MtCO_{2eq}$, more than half of which (114.7 $MtCO_{2eq}$) originated from soy exported to China. Land use change accounted for the largest share of total GHG emissions of exported soy (74.81 $MtCO_{2eq}$), followed by domestic transportation (57.89 $MtCO_{2eq}$) and processing stage (46.03 $MtCO_{2eq}$). The remaining emissions came from agricultural production and international transportation [89].

Saget et al. (2020) applied LCA to compare different environmental impacts of pasta made from chickpea with conventional pasta from durum wheat, following an approach from cradle to fork [90]. Two FUs were selected: one serving being equivalent to 80 g of dry pasta, and one nutrient density unit. The nutrient density unit considered the amounts of the most essential nutrients including essential fatty acids, protein, and fiber in the food product, which is 2.3 for cooked chickpea pasta and 0.9 for cooked wheat pasta. The environmental burden per serving was smaller for chickpea pasta across at least 10 of 16 impact categories examined. When quantified per nutrition density unit, the environmental advantage of chickpea pasta extended to 15 of the 16 categories. The only exception was the land use burden of chickpea pasta, which was 17% higher than that of conventional pasta. The hotspot of the impact was the agriculture stage, accounting for at least 20% of the total impacts in 15 out of 16 impact categories [90].

Saget et al. (2021) also conducted a LCA on pea protein ball and beef meatball from cradle to fork (excluding packaging and recycling) [91]. The FUs of one serving and one nutrient density unit were selected. In this study, one serving was equivalent to 100 g of cooked pea protein ball or beef meatball. Pea protein ball was associated with lower environmental burdens across all 16 impact categories per serving, as well as per nutrition density unit. GWP of pea protein ball was 0.5 kg $CO_{2eq}$ per serving, at least 85% smaller than that of beef meatball. Other impact categories such as AP and land use of pea protein ball ($3.4 \times 10^{-3}$ mol $H^+_{eq}$ and 38 points, respectively) were also significantly lower than that of beef meatball per serving. Similar results were obtained in the relative variance between the GWP (85% lower), AP (87% lower), and land use (93% lower) of pea protein ball and beef meatball per nutrient density unit [91].

Costantini and Bacenetti (2021) investigated the environmental impacts of maize and soybean in Paraguay [92]. Two intra-annual rotations were examined, in which the first season crop is soybean, and the second season crops are maize or soybean. The product systems were assessed from cradle to farm gate, excluding long-term land impacts due to short cropping system and to the fact that land use change would not affect the relative comparison between two cropping systems. A part from the mass-based FU (one tonne of product, i.e., maize or soybean as either first or second season crop), several other FUs were applied, including land management FU (1 ha per year), productive FU (1 GJ of gross energy per ha per year and 1 t of crude protein per ha per year), and financial FU (1 USD of gross margin per ha per year) [92].

It was identified that, per tonne of product, the first season soybean's GWP was 284.4 kg $CO_{2eq}$, higher than that of second season maize (182.4 kg $CO_{2eq}$) but lower than that of second season soybean (590.8 kg $CO_{2eq}$). GWP of the soybean–maize cropping system was 1183.3 kg $CO_{2eq}$ per tonne of crude protein per ha per year, similar to that of the soybean–soybean rotation (1179.2 kg $CO_{2eq}$). Other impact categories such as ODP, fine particular matter formation, terrestrial acidification, mineral resource scarcity, and fossil resource scarcity were similar for both cropping systems [92].

Freshwater eutrophication (FEP) of the soybean–soybean cropping system was 1.41 kg $P_{eq}$, triple that of the soybean–maize cropping system, at 0.4 kg $P_{eq}$ per one tonne of crude protein per ha per year. In contrast, FETP of soybean–maize cropping system was 71.2 kg $1,4DCB_{eq}$, much higher than that of the soybean–soybean cropping system, at 53.1 kg $1,4DCB_{eq}$ per one tonne of crude protein per ha per year. The soybean–maize cropping system was more efficient in terms of land management, gross energy, and gross margin per ha per year. For example, GWP of the soybean–maize cropping system per ha per year was 1871.4 kg $CO_{2eq}$, which is slightly lower than that of the soybean–soybean cropping system, at 1882.3 kg $CO_{2eq}$. Per USD 1 of gross margin per ha per year, GWP of soybean-maize cropping system was 1.46 kg $CO_{2eq}$, lower than that of the soybean–soybean cropping system (1.73 kg $CO_{2eq}$) [92].

Järviö et al. (2021) conducted a cradle-to-gate LCA on microbial protein produced by autotrophic hydrogen oxidizing bacteria in the context of a factory in Finland [93]. Environmental impacts of microbial protein, assessed per one kilogram of product, were compared with animal and plant-based protein for food and feed production. The assessed environmental indicators include GWP, land use, FEP, water scarcity, human (non) carcinogenic toxicity, and cumulative energy demand. GWP of microbial protein was 8.38 kg $CO_{2eq}$, with Finish average energy mix. This impact was much lower when the hydropower was used in replacement of average energy mix (1.04 kg $CO_{2eq}$). FEP was $2.36 \times 10^{-3}$ kg $P_{eq}$, and the FETP was 0.33 kg $1,4DCB_{eq}$ per kg of microbial protein. Compared to animal-based protein sources for food production, microbial protein had lower environmental impacts. For example, the GWP of microbial protein produced with Finnish average energy mix equals 6.2% of GWP of bovine meat protein from beef herd and 7.3% of that from dairy herd. Compared to plant-based protein sources for food production, microbial protein had lower land and water use requirements and EP, but GWP only reduced if low emission energy sources (for example hydropower) were used [93].

Üçtuğ et al. (2021) assessed the life cycle impacts of weekly omnivorous, vegetarian, and vegan diets based on traditional Turkish cuisine from cradle to grave [94]. The authors assumed that all populations could receive 2000 kcal daily intake per person. It was identified that the GWP of omnivorous, vegetarian, and vegan diets were 35.22, 27.8, and 18.5 kg $CO_{2eq}$, respectively. Most of the impacts came from the raw material supply, contributing to 77.2% of overall impact, followed by meal preparation, which accounted for 21.5% [94].

Lie-Piang et al. (2021) assessed the environmental impacts of protein concentrates of crops obtained with different technologies, including conventional fractionation, mild aqueous fractionation, dry fractionation, and combined fractionation [95]. The system boundaries covered the crop cultivation, transportation, and processing stage. It was identified that conventional fractionation caused the highest impact on GWP (1509 kg $CO_{2eq}$/tonne of crop or 5.33 kg $CO_{2eq}$/kg of protein for yellow pea and 2106 kg $CO_{2eq}$/tonne of crop or 5.78 kg $CO_{2eq}$/kg of protein for lupine). Dry fractionation is responsible for the lowest impact on GWP: 676 kg $CO_{2eq}$/tonne of crop or 1.58/Kg of protein for yellow pea, and 750 kg $CO_{2eq}$/tonne of crop or 1.3 kg $CO_{2eq}$/kg of protein for lupine. A large relative difference was observed between GWP of mild aqueous and conventional fractionation of yellow pea (about three times) per kg of protein, while the relative difference between the same technologies per tonne of crop is small (about 1.3 times) due to a higher protein yield of mild aqueous fractionation. In contrast, the relative difference between GWP of combined fraction and conventional fraction per tonne of crop and that of the same

technologies per kg of protein are both about two times, indicating that the combined fractionation considerably reduced environmental impact as well as protein yield [95].

FEP was similar among technologies, ranging from 0.37 to 0.54 kg $P_{eq}$ per tonne of crop. Land use varied from 3463 $m^2$a $crop_{eq}$ for mild aqueous fractionation of yellow pea to 3517 $m^2$a $crop_{eq}$ for conventional fractionation, dry fractionation and combined fractionation of yellow pea, and 6464 $m^2$a $crop_{eq}$ for lupin regardless of technologies [95].

Tidåker et al. (2021) evaluated the environmental impacts of five Swedish pulses, including yellow peas, grey peas, faba beans, common beans, and lentils from cultivation to factory gate [96]. Yellow peas, faba beans, and common beans were single crops, while grey peas, and lentils were grown with oats. The results of the analysis showed that the intercropping pulses and cereals showed potential to reduce environmental pressures. In detail, per kg of dry product, the energy and GWP impacts varied from 1.6 MJ (for organic grey pea) to 3.3 MJ (for conventional common bean) and from 0.18 kg $CO_{2eq}$ (for organic grey pea, conventional faba bean and conventional yellow pea) to 0.44 kg $CO_{2eq}$ (for conventional common bean), respectively. GWP per kg of cooked product ranged from 0.1 kg $CO_{2eq}$ for pulses purchased dry to 0.8 kg $CO_{2eq}$ for canned beans. Long transport distances contributed considerably to energy use and climate impact, particularly when the pulses were processed and packaged far from the final destination due to the high moisture content of the product [96].

### 3.3. Methodological Aspects

Different methodological aspects regarding the FUs, system boundaries, allocation procedures, data acquisition and data quality, and characterizing the reviewed case studies are examined in the following sub-sections.

### 3.3.1. Multiple Functional Units

One of the most common FUs selected for agricultural products is a mass-based FU (e.g., kg of product), as it is widely accepted that the main function of a single crop is to deliver a certain quantity of a product [76,80–93,95,96].

However, the function of agricultural products does not limit to their quantity, but it can include the function to provide an energy value, nutrition value, protein value, etc. The question of a suitable FU arises in the case the LCA study, which aims to compare several products of different qualities and of different nutrient composition. In this case, the application of mass-based FU can cause a lack of transparency, comprehensiveness, and consistency. These drawbacks of mass-based FU induce the need of a novel FU. Some authors proposed nutrient-related FU to make it feasible to compare several food products with different nutrient qualities. For example, the FU of protein content was applied in several studies such as [77,86,87,92,95]. Another nutrient-related FU is based on energy content, for example, 3.75 MJ energy content with corrected weight in [78] and GJ per ha per year in [92]. Additionally, the FU of a meal, diet, or daily intake was employed and frequently converted into kcal of energy, which were used in five case studies [75,79,85,88,94]. Refer to Table 2 for detailed FUs.

Due to the complexity of the nutrients contained in the food, some authors proposed a FU composed of multiple nutrients. The nutrient index, for example, is the combination of macronutrients, micronutrients, and limited nutrients at the same time [97]. In detail, Saarinen et al. (2017) employed six nutrient-based FUs [97]. The first two FUs were based on the Nutrient Rich Food index, including several recommended nutrients of protein, fiber, Ca, Fe, Mg, K, vitamin A, C, E, Na, and limited nutrients of saturated fatty acids and added sugar. Two other FUs were based on the Finnish Nutrient Index, considering protein, fiber, mono-unsaturated fatty acids, poly-unsaturated fatty acids, Ca, Fe, vitamin B2, folic acid, Na, saturated fatty acids, and added sugar. The last two FUs took into account only limited nutrients such as Na, saturated fatty acid, and added sugar [97].

Other authors studied the most essential nutrients, for example, the nutrient density unit, which was calculated based on essential fatty acid, protein, and fiber [90,91]. The nutrient density unit, which was proposed by van Dooren (2016), showed the relation between the amounts of encouraged macronutrients (essential fatty acids, protein, and fiber) in the food and the total energy provided by the food for 100 g of food product [98]. This unit was converted into a dimensionless one by dividing the amounts of encouraged macronutrients by their recommended amounts (g) and dividing the food's total energy by the recommended daily energy value (kcal). The higher nutrient density unit means the higher amount of encouraged macronutrients that the food would provide. The nutrient density units of food are various: 1 for semi-skimmed milk, 2 for raw pork, and 2.8 for canned pulses [98].

In some cases, mass-based FU and nutrient-related FU were simultaneously employed in the same study. Smetana et al. (2015) used four different FUs: one kilogram of ready-to-eat meal at a consumer (first FU), equivalent to 3.75 MJ energy content (second FU), one kilogram of product (third FU), and one kilogram of product with corrected weight (forth FU) [78]. Costantini and Bacenetti (2021) used several FUs for fulfilling different function. For the function of supplying a product for general use, the authors used one tonne of individual product at 14% moisture content. For the function of land management, the authors employed ha per year. Two FUs of GJ per ha per year and one tonne of crude protein per ha per year were utilized to indicate the productive function. Finally, for the financial function, USD of gross margin per ha per year was applied [92].

### 3.3.2. System Boundaries

Overall, 13 out of 22 case studies considered the product system from cradle to farm or factory gate [76,80–84,86,87,89,92,93,95,96]. The selection of the cradle to farm or factory gate system boundaries can deal with the similarity among the studied product systems in terms of environmental impacts from cooking procedure during the consumption and waste management during the end-of-life of the food products. Additionally, this may originate from the difficulty in modelling the different consumption patterns of food. There are five case studies considering the impacts of food cooking and food consumption, or the cradle to plate or fork system boundaries [77,88,90,91,97]. The number of cradle-to-grave LCA studies is four [75,79,85,94]. A summary of selected system boundaries is reported in Table 2.

**Table 2.** Functional units and system boundaries of reviewed LCA case studies.

| No. | Paper | Topic | Functional Unit | System Boundary |
|---|---|---|---|---|
| 1 | Davis et al. (2010) [75] | Soybean and pea | A meal served at the household | Cradle to grave |
| 2 | Knudsen et al. (2010) [76] | Soybean | One tonne of soybean | Cradle to farm gate |
| 3 | Zhu and Ierland (2004) [77] | Pork vs. dry pea | 1000 kg of protein content | Cradle to plate |
| 4 | Smetana et al. (2015) [78] | Plant/animal-based diet (chicken, dairy-based, lab-grown, insect-based, gluten-based, soy meal-based, and mycoprotein-based products) | One kilogram of ready-to-eat meal at consumer 3.75 MJ energy content with corrected weight | Cradle to plate |
| 5 | Sturtewagen et al. (2016) [79] | Canteen/home pork vs. quorn (six meals) | A meal | Cradle to grave |
| 6 | Lathuillière et al. (2017) [80] | Soybean | One tonne of soybean | Cradle to farm gate |
| 7 | Mierlo et al. (2017) [81] | Plant/animal-based diet (chicken and beef, vegetarian: plant + animal origin—insect vegan: totally plant-based insect-based fortification-free) | One kilogram of meat replacer | Cradle to farm/factory gate Ingredient system boundary (from agricultural production to factory processing raw material into ingredients, for example, from soybean to soy flour or from chicken breeding to egg) End-product system boundary (from agricultural production to factory processing ingredients into meat replacer, for example, from soybean to chicken/beef replacers) |
| 8 | Cancino-Espinoza, et al. (2018) [82] | Quinoa | 500 g package of organic quinoa | Cradle to distributors |
| 9 | Corrado et al. (2019) [85] | Plant/animal-based diets (omnivorous, vegetarian and vegan) | One kilogram of edible food Diet to meet nutrient requirement of an average Italian man | Cradle to grave |
| 10 | Ilari et al.,(2019) [84] | Frozen green bean | One kilogram of frozen green bean | Cradle to factory gate |
| 11 | Lee and Choe (2019) [83] | Soybean | One tonne of soybean | Cradle to farm gate |
| 12 | Escobar et al. (2020) [99] | Soy (bean, oil, and protein cake) | One tonne of product | Cradle to factory gate |
| 13 | Heusala et al. (2020a) [86] | Oat protein concentrate | One kilogram of ready-to-eat product One kilogram of protein | Cradle to factory gate |

**Table 2.** *Cont*.

| No. | Paper | Topic | Functional Unit | System Boundary |
|---|---|---|---|---|
| 14 | Heusala, et al. (2020b) [87] | Oat and faba bean protein concentrate | One kilogram of ready-to-eat product One kilogram of protein | Cradle to factory gate |
| 15 | Mogensen et al. (2020) [88] | Oat protein concentrate | One kilogram of ready-to-eat product Daily intake of food and beverages for one adult | Cradle to plate |
| 16 | Saget et al. (2020) [90] | Wheat vs. chickpea pasta | 80 g of dry weight pasta Nutrient density unit | Cradle to fork |
| 17 | Costantini and Bacenetti (2021) [92] | Soybean and maize | One tonne of individual product Ha per year, GJ per ha per year, tonne of crude protein per ha per year, USD of gross margin per ha per year | Cradle to farm gate |
| 18 | Järviö,et al. (2021) [93] | Microbial protein | One kilogram of microbial protein product | Cradle to gate |
| 19 | Lie-Piang et al. (2021) [95] | Protein concentrate of oil and starch bearing crops | One tonne of the processed crop One kilogram of protein in the produced fraction | Cradle to factory gate, including processing and crop cultivation |
| 20 | Saget et al. (2021) [91] | Beef vs. pea protein | 100g serving of cooked protein balls | Cradle to fork, excluding packaging and recycling |
| 21 | Tidåker et al. (2021) [96] | Yellow peas, grey peas, faba beans, common beans, and lentils | One kilogram of dry product | Cradle to factory gate, from cultivation, processing, packaging, and transport |
| 22 | Üçtuğ et al. (2021) [94] | Plant/animal-based diets (omnivorous, vegetarian, and vegan diet) | 2000 kcal of daily intake per person | Cradle to grave |

### 3.3.3. Allocation Methods

Among 22 case studies, there were four papers presenting no information on allocation and examining single-output processes [79,82,83,93]; the 18 remaining papers either applied no allocation, mass allocation, or economic allocation. Six case studies, including [76,80,84,85,92,94], did not require allocation in their studies, as there were no co-products or by-products in the studied product systems. In some of the above cases, the crop residues were considered waste due to the difficulty in ascribing them an economic value. Mass allocation was applied in four case studies [78,89,95,96]. In these case studies, the allocation was based on weight, dry matter, or kilograms of protein content of products and co-products. Economic allocation was used in five papers [77,81,87,88,90].

Three case studies applied no allocation, economic allocation, and mass allocation in combination. In comparative studies such as those of Davis et al. (2010) and Heusala et al. (2020a), economic allocation and no allocation were simultaneously applied [75,86]. Davis et al. (2010) compared the environmental burdens of meals with pork chops, pea-meat sausage, and pea burger and applied economic allocation for crop-based ingredients, e.g., wheat flour and its co-product (bran) to make wheat bread used in all meals. Additionally, no allocation was required for pork, which was used in pork chop meals and pea-meat sausage meals [75]. Heusala et al. (2020a) compared regular food products and food products containing OPC. For oat products, environmental impacts were allocated based on their economic value, whereas no allocation was applied for regular food products, including wheat flour and durum semolina [86]. In Saget et al.'s study (2021), both economic and physical allocations were applied for pea and its co-products during the dehulling and fractionation processes [91]. For beef products, both economic and biophysical allocation were performed [91]. The biophysical allocation was proposed by Chen et al. (2017) for assessing life cycle environmental impact of meat products. It was based on metabolic energy requirement for the animal growth from its birth to the slaughter age [100]. The application of different allocation methods highlighted their roles in assessing life cycle environmental impacts of agricultural products, in which the environmental impacts of beef meatball considerably reduced when the biophysical allocation method was applied [91].

### 3.3.4. Data Acquisition and Quality

Data of reviewed case studies were obtained from a mixture of sources, such as directly obtained from farmers and manufacturers or indirectly extracted from inventory databases, modelling, and the literature. Overall, 14 out of 22 case studies obtained primary data for foreground processes through interviews, questionnaires, and surveys. Data for background process were taken from the literature and inventory databases. The data quality met the requirements on representativeness, completeness, precision, and methodological appropriateness and consistency.

The remaining case studies depended on the literature and inventory databases for data collection. It should be noted that the product systems of these case studies are either a diet/meal (with mixtures of ingredients) or general products, for which it is difficult and unfeasible to collect on-site data.

The life cycle inventory database, Ecoinvent, with different versions (https://www.ecoinvent.org) (accessed on 30 November 2021), was used in 13 case studies. Additionally, there are some food life cycle inventory databases, such as Agri footprint (https://www.agri-footprint.com) (accessed on 30 November 2021), LCA food DK (http://www.lcafood.dk/) (accessed on 30 November 2021), and less common life cycle databases such as Gabi database, EDP database (www.environdec.com) (accessed on 30 November 2021), SEI-PCS database (https://trase.earth) (accessed on 30 November 2021), and Global Soil Dataset for Earth modelling (http://globalchange.bnu.edu.cn) (accessed on 30 November 2021), have been employed.

### 3.4. Obtained Results: Environmental Impacts and Comparison with Meat

One of the most common environmental impact indicators is climate change (or GWP or GHG emissions), which was assessed by 20 out of 22 case studies. Other impact indicators that gather a lot of attention are land use (15 case studies), energy consumption or energy efficiency (14 case studies), and water use or water impact (12 case studies).

Table 3 presents the obtained results on GWP of different high-protein food to replace meat in the reviewed case studies.

**Table 3.** Obtained results on GHG emissions in some reviewed LCA case studies.

| Paper | GWP (kg CO$_{2eq}$) | Per kg of Product | Per kg of Protein | Per Meal/Diet [3] |
|---|---|---|---|---|
| Cancino-Espinoza et al. (2018) [82] | Quinoa<br>Rice | 0.88 | 6.47<br>34.1 | |
| Costantini and Bacenetti (2021) [92] | Maize | 0.18 | | |
| Zhu and Ierland (2004) [77] | Dry pea | | 12.23 | |
| Ilari et al. (2019) [84] | Frozen green bean | 0.7 | | |
| Escobar et al., (2020) [89] | Soybean (Brazil) | 0.69 | | |
| Costantini and Bacenetti (2021) [92] | Soybean | 0.28–0.59 | | |
| Smetana et al. (2015) [78] | Soy meal product | 2.65–2.78 | | |
| Lee and Choe (2019) [83] | Conventional soybean (Korean)<br>Organic soybean (Korean) | 1.65<br>2.04 | | |
| Knudsen et al. (2010) [76] | Conventional soybean (at Chinese farm gate)<br>Organic soybean (at Chinese farm gate)<br>Organic soybean (from China imported to Denmark) | 0.26<br>0.15<br>0.42 | | |
| Tidåker et al. (2021) [96] | Dry pulses (Sweden)<br>Cooked pulses at home (Sweden)<br>Canned beans (Sweden) | 0.18–0.44<br>0.1<br>0.8 | | |
| (Heusala et al., 2020a) [86] | Wheat flour<br>Durum wheat semolina<br>Oat starch<br>Oat protein concentrate | 0.5<br>1<br>1<br>3.3 | 4.8<br>7.8<br>12.4<br>8.8 | |
| Heusala et al. (2020b) [87] | Faba bean protein concentrate | 1.1–2 | 1.9–3.4 | |
| Lie-Piang et al. (2021) [95] | Yellow pea protein concentrate<br>Lupine protein concentrate | 0.67–1.5<br>0.75–2.1 | 1.58–5.33<br>1.3–5.78 | |
| Smetana et al., (2015) [78] | Mycoprotein based product | 5.55–6.15 | | |
| Järviö et al. (2021) [93] | Microbial protein product | 8.38 | | |
| Saget et al. (2020) [90] | Wheat pasta<br>Chickpea pasta (Bulgaria)<br>Chickpea pasta (Spain)<br>Cooked pea protein ball | 2.45–2.58<br>2.03<br>1.42<br>0.5–1.1 | | |

**Table 3.** *Cont.*

| Paper | GWP (kg CO$_{2eq}$) | Per kg of Product | Per kg of Protein | Per Meal/Diet [3] |
|---|---|---|---|---|
| Mierlo et al. (2017) [81] | Vegetarian/vegan ingredient product in replacement of chicken | 0.62–1.35 | | |
| | Vegetarian/vegan ingredient product in replacement of beef | 0.59–1.31 | | |
| | Vegetarian/vegan end product in replacement of chicken | 0.62–0.73 | | |
| | Vegetarian/vegan end product in replacement of beef | 0.59–0.7 | | |
| Üçtuğ et al. (2021) [94] | Vegetarian diet | | | 3.97 |
| | Vegan diet | | | 2.64 |
| Corrado et al. (2019) [85] | Vegetarian | | | 2.76–3.2 |
| | Vegan | | | 2.61–3.13 |
| Mogensen et al. (2020) [88] | Diet with OPC bread, pasta, and oatgurt | | | 3.95 |
| | Diet with OPC in replacement of meat | | | 3.34 |
| Davis et al. (2010) [75] | Pea sausage (Sweden) | | | 1.22 |
| | Pea burger (Sweden) | | | 0.54 |
| | Pea sausage (Spanish) | | | 1.74 |
| | Pea burger (Spanish) | | | 1.16 |

[3] Meals and diets of different studies provide various energy value. Specifically, they provide 2000 kcal per day in [94], 2131 kcal per day in [85], 2107–2389 kcal per day in [88], and 750–780 kcal in [75].

There is considerable variance among the GWP of high-protein products, ranging from 0.1 kg CO$_{2eq}$ of Swedish pulse cooked at home [96] to 8.38 kg CO$_{2eq}$ of microbial protein product [78]. Even among the same product, the difference in the obtained results of different case studies is significant. The GWP of soybean, for example, ranges from 0.1 to 2.78 kg CO$_{2eq}$ per kg of soybean. This difference originates from the specific soybean product (organic or conventional), country of origin (Brazil, Korean, China), and studied system boundaries (cradle to farm gate or cradle to plate).

Additionally, there is a relative difference among the impacts when applying different FUs. In detail, in the case studies that applied several FUs cases, the GWP increases when it is assessed per kg of protein. The changes in the impacts when being assessed per different FUs is acceptable if the case studies are conducted for one product. However, if the case study is a comparative LCA, the choice of FU is important [101–103]. The study of Heusala et al. (2020a) identified that the GHG emissions of durum wheat semolina is 1 kg CO$_{2eq}$ per kg of product, which equals those of oat starch. However, when being assessed per kg of protein, the GHG emissions of oat starch are nearly double those of durum wheat semolina (12.4 kg CO$_{2eq}$ vs. 7.8 kg CO$_{2eq}$) [86].

The land use impact of high-protein food to replace meat is quantified in m$^2$ or m$^2$ per year or point of land use or m$^2$a of land occupation per FU. A significant variance in land use impact has been observed, in which the lowest is 0.37 m$^2$/kg of cooked oat starch pasta [86], and the highest is 34.7 m$^2$/kg of protein from FBC [87]. This large variance originates from the difference in the applied FU. In most cases, the land use impact is smaller when being quantified per kg of product, the range being between 0.37 m$^2$/kg of cooked oat starch pasta, as in [86], and 5.9 m$^2$/kg of dry bean, as in [96]. Meanwhile, per kg of protein, the land use impact varies from 8.6 m$^2$/kg of OPC [86] to 34.7 m$^2$/kg of FBC [87].

Land occupation was evaluated for soybean [80], yellow pea, lupine [95], and microbial protein [93]. It was identified that land occupation of microbial protein product was the smallest, at 0.3 to 0.7 m$^2$a crop$_{eq}$, depending on different energy mixes [93]. The result is reasonable, as the microbial protein is completely manufactured in the laboratory and does not require any agricultural land. Meanwhile, the land occupation of soybean and

yellow pea were considerably higher, at 2.93 m$^2$a/kg of soybean grown in the Amazon [80], 3.5 m$^2$a crop$_{eq}$ per kg of yellow pea, and 6.4 m$^2$a crop$_{eq}$ per kg of lupine [95].

The units of m$^2$ per year and point of land use impact are less common than the two units mentioned above. Specifically, the unit of m$^2$ per year was only applied in Mierlo et al. (2017) to quantify land use of ingredients and end-products to replace chicken and beef [81]. Land use of vegetarian and vegan ingredients to replace chicken and beef range from 2.52 to 6.51 m$^2$/year/kg, while those of end products range from 2.53 to 3.32 m$^2$/year/kg. Similarly, one group of authors used the unit of point of land use impact [90,91]. In Saget et al. (2020), land use impact was quantified for Spanish and Bulgarian chickpea pasta (at 32.9 to 36.7 points per 80 g of serving and 60.6 to 67.4 points per nutrient density unit) [90]. Later on, Saget et al. (2021) quantified land use impact of cooked pea protein ball, which was 38 points per 100 g of serving and 20.8 points per nutrient density unit [91].

Regarding energy consumption, two common indicators are selected including renewable and non-renewable energy consumption. These indicators are quantified in either MJ or kg oil$_{eq}$. For example, energy consumption of soybean was lowest at 0.77 MJ per kg of soybean [76] and highest at 36.9 MJ/kg of soymeal product [78]. While Knudsen et al. (2010) quantified the energy consumption from cradle to farm gate [76], the study of Smetana et al., (2015) covered from cradle to plate [78]. The difference in studied system boundaries, whether covering energy intensive stages such as processing and cooking, may cause a significant range in the obtained results of energy consumption. While the unit of MJ can be applied for both renewable and non-renewable energy consumption, the unit of kg oil$_{eq}$ was used for assessing non-renewable energy consumption. For example, in the study of Lie-Piang et al. (2021), depending on the technologies, the fossil resource scarcity of yellow pea was 104 to 326 kg oil$_{eq}$/tonne of crop, and that of lupine was 128 to 427 kg oil$_{eq}$/tonne of crop [95].

The indicators in terms of water impact of high protein food to replace meat are diverse with FEP, FETP and water consumption. Specifically, FEP results range between 0.13 kg P$_{eq}$/tonne of soybean [92] to 0.54 kg P$_{eq}$/tonne of lupine [95]. Meanwhile, the FETP results are variant from 14 kg 1,4DCB$_{eq}$/tonne of soybean [92] to 334 kg 1,4DCB$_{eq}$/tonne of microbial protein product [93]. Water consumption ranges between 10 m$^3$/tonne of dry pea [77] to 34 m$^3$/tonne of yellow pea [95], with the exceptional lowest water consumption of lupine with dry fraction technology (at 2.8 m$^3$/tonne of lupine) and the exceptional highest water consumption of microbial protein product (at 155–198 m$^3$/tonne of product, depending on energy mixes).

In general, high-protein food of plant origin is a good choice in term of environmental impacts, compared to meat. Among the most common impact categories, including climate change, land use, energy consumption, and water impacts, plant-based high-protein food has environmental advantages. Specifically, the climate change impact of pork chop, either fed with soy or pea, ranged from 1.15 to 1.77 kg CO$_{2eq}$, which was higher than that of sausage made from pea and meat, or pea burger, at 0.54 to 1.74 kg CO$_{2eq}$ [75]. Similarly, the vegetarian- and vegan-based chicken and beef replacers emitted 0.59–1.31 kg CO$_{2eq}$ per kg of product, much lower than the emissions of chicken and beef, at 2.5 and 7.53 kg CO$_{2eq}$, respectively [81].

Other impact indicators, such as land use and water use show similar patterns. Specifically, land use per one tonne of pork was 5.5 ha, which was three times higher than that of alternative protein food from dry pea. Water use of pork was 36 thousand m$^3$ per tonne, which was much higher than 10.9 thousand m$^3$ per tonne of alternative protein food [77]. Land use of chicken and beef was 4.7 and 5.8 m$^2$/year per kg of product, respectively. These numbers were higher than the average land use of chicken vegetarian/vegan replacer, ranging from 1.64–6.51 m$^2$/year/kg, and beef vegetarian/vegan replacer, ranging from 2.52–6.32 m$^2$/year/kg [81]. At the same time, water consumption of chicken and beef was 0.05 and 0.06 m$^3$ per kg of product, respectively. These numbers were higher than the average water use of vegetarian/vegan chicken replacer, ranging between

0.017 and 0.069 m$^3$/kg, and vegetarian/vegan beef replacer, ranging between 0.019 and 0.07 m$^3$/kg [81].

Additionally, per nutrient density unit, the land use and water scarcity of plant-based high-protein food is smaller than meat-based food. For example, the land use of pea protein ball is 20.8 points, which is considerably lower than that of beef meatball, at 266.3–891.4 points (depending on origin of the meat) per nutrient density unit. At the same time, the water scarcity of pea protein ball is two to five times lower that of beef meatball (at 0.14 vs. 0.32–0.78 m$^3$ of water deprivation per nutrient density unit) [91].

Regarding energy consumption, plant-based high-protein food consumes less fossil fuels. The vegetarian and vegan-based chicken replacer required 7.28–15.78 MJ per kg of product compared to 17 MJ for one kilogram of chicken. Furthermore, the vegetarian- and vegan-based beef replacer required 6.78–15.27 MJ per kg of product, compared to 21 MJ for one kilogram of beef [81]. When comparing the energy consumption of plant-based high-protein food and conventional food as meals, including energy consumption for preparing the ingredients, cooking the meals, and waste management, there was an insignificant difference between pork and plant-based high-protein alternatives (13–22.4 MJ$_{eq}$ for pork chop feed with soy, 12.9–22 MJ$_{eq}$ for pork chop feed with pea, 13.2–20.4 MJ$_{eq}$ for pea-meat sausage, and 11.2–17.7 MJ$_{eq}$ for pea burger) [75]. In all cases, the plant-based high-protein alternatives consumed smaller amounts of energy.

### 4. Conclusions

The literature review on the life cycle impacts of plant-based high-protein food that can be used as meat replacers indicated that the environmental impacts are variable among specific high-protein foods. This occurs in the four most commonly assessed indicators, including GWP, land use, energy consumption, and water impacts. The significant variance in the life cycle environmental impacts of these high-protein foods originates from the nature of specific product systems (some are grown on the field, and others are made in the laboratory), countries of origin, FUs, and system boundaries. Despite the significant variance among the plant-based high-protein foods, they show better environmental results than meat-based products in all four commonly assessed environmental indicators. Even when being quantified with different FUs such as a meal or one nutrient density unit, the plant-based high-protein food is more environmentally friendly than meat. Moreover, previous research points to additional health benefits of vegetarian or flexitarian nutrition. However, current interventions do not take into account the concept of non-inferiority. Design and interpretation of such trials is more complex and rarely used as standard method to test efficacy between nutrition from animal and plant-based sources. As, therefore, many studies are inconclusive, it can be concluded that the consumption of meat and meat products can be drastically reduced to general recommendation levels of 350–500 g/week without having to fear negative health consequences.

Regarding the methodological aspect of LCA, the FU is diversified by mass-based and nutrient-related FU. Additionally, it is common to apply several FUs in one study in order to present distinct functions of the food product system. The popular system boundaries are from cradle to farm or factory gate. This may be due to the difficulty in modelling the consumption and end-of-life of food product system and the similarity of cooking, eating patterns, and waste treatment among studied food products. The frequently applied allocation methods are based on mass and economic values. In terms of data acquisition and quality, the EcoInvent database was used in more than half of the case studies. Additionally, other food relevant databases such as Agri footprint and LCA food DK were utilized.

**Author Contributions:** Conceptualization, all authors; Methodology, S.L. and L.Q.L.; Software, M.A.C. and M.C.; Validation, M.C. and T.S.; Formal analysis, L.Q.L.; Investigation, M.A.C., S.L. and L.Q.L.; Resources, M.C. and T.S.; Data curation S.L., L.Q.L. and T.S.; Writing—original draft preparation, M.A.C., L.Q.L. and T.S.; Writing—review and editing, S.L. and L.Q.L.; Supervision, M.C. All authors have read and agreed to the published version of the manuscript.

**Funding:** This research was funded within the call ERA—HDHL—KH_FNS Knowledge Hub on Food and Nutrition Security. The APC was funded by the German Federal Ministry of Food and Agriculture (BMEL) through the Federal Office for Agriculture and Food (BLE), grant number 2819ERA15E.

**Institutional Review Board Statement:** Not applicable.

**Informed Consent Statement:** Not applicable.

**Data Availability Statement:** Not applicable.

**Acknowledgments:** This research was developed within the project "SYSTEMIC: An integrated approach to the challenge of sustainable food systems: adaptive and mitigatory strategies to address climate change and malnutrition", funded within the call ERA—HDHL—KH_FNS Knowledge Hub on Food and Nutrition Security.

**Conflicts of Interest:** The authors declare no conflict of interest. The funders had no role in the design of the study; in the collection, analyses, or interpretation of data; in the writing of the manuscript, or in the decision to publish the results.

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
