# Peer review of "Life Cycle Environmental Impacts and Health Effects of Protein-Rich Food as Meat Alternatives: A Review"

_sustainability, doi:10.3390/su14020979_

Round 1

Reviewer 1 Report

In this review, titled: "Life cycle environmental impacts and health effects of protein-rich food alternative to meat" authors gave an overview of  the health aspects of protein-rich food alternative to meat and carry out a literature review on the life-cycle environmental impacts of this alternative food. The document is well written and interesting, only a few revisions are suggested: 

  • In my opinion, the title should contain the word "Review" to make the manuscript type easier to understand;
  • The keywords should not overlap with the title, it is recommended to change them;

  • There is a reference in the text missing the year (Saget et al. ??)

  • Line 604: grams must be indicated as "g" and not "gr".  

Author Response

Response to the Reviewers

The authors would like to thank the Reviewers for their thorough analysis that will for sure help in improving the scientific quality of the paper. In the following pages, the comments of the Reviewers are reported and, for each comment, a reply from the authors is included in the text.

Reviewer 1: In this review, titled: "Life cycle environmental impacts and health effects of protein-rich food alternative to meat" authors gave an overview of the health aspects of protein-rich food alternative to meat and carry out a literature review on the life-cycle environmental impacts of this alternative food. The document is well written and interesting, only a few revisions are suggested.

Authors: Thank you for the appreciation. Authors modified the paper according to your suggestions and comments.

 Reviewer 1: In my opinion, the title should contain the word "Review" to make the manuscript type easier to understand.

Authors: According to your suggestion the title was modified as follows: Life cycle environmental impacts and health effects of protein-rich food as alternative to meat: a review.

Reviewer 1: The keywords should not overlap with the title, it is recommended to change them.

Authors: Thank you for this comment. The keywords are changed as follows: life cycle assessment, proteins, food, environmental sustainability, human health.

Reviewer 1: There is a reference in the text missing the year (Saget et al. ??)

Authors: Thank you for highlighting this missing information, which was added in the text: “In Saget et al.’s study (2021) […]”.

Reviewer 1: Line 604: grams must be indicated as "g" and not "gr".  

Authors: Thank you for your suggestion. Throughout the text, “gr” was substituted with “g”.

Reviewer 2 Report

Reviewer’s Report

MS ID: 1560178

Title: Life cycle environmental impacts and health effects of protein-rich food alternative to meat

In the title change the alternative to alternativeS

Abstract: 1. Rewrite it more clearly to attract a greater number of specific and non-specific readers

Introduction:

  1. In the last paragraph rewrite the objective of the review clearly.
  2. Line no. 105-107 confusing statement.
  3. Line No. 131-134 what it mean?
  4. Need to present more data on specific meat alternative plant-based protein sources.
  5. The LCA methods were well discussed and compared the published methods.

Author Response

Response to the Reviewers

The authors would like to thank the Reviewers for their thorough analysis that will for sure help in improving the scientific quality of the paper. In the following pages, the comments of the Reviewers are reported and, for each comment, a reply from the authors is included in the text.

Reviewer 2: In the title change the alternative to alternativeS

Authors: Thank you for your suggestion. However, the intention of the authors is to refer to use the singular “alternative” as adjective of the word food. However, in order to avoid confusion in the reader, the title was modified as follows: Life cycle environmental impacts and health effects of protein-rich food as alternative to meat: a review.

Reviewer 2: Abstract: Rewrite it more clearly to attract a greater number of specific and non-specific readers

Authors: According to your suggestion, the abstract was modified as follows: “The food sector is responsible of considerable environmental impacts in most environmental contexts: the food supply chain causes greenhouse gas emissions, water consumption, reduction of cultivable land and other environmental impacts. Thus, a change in food supply is required to reduce the environmental impacts caused by the food supply chain and to meet the increasing demand for sufficient and qualitative nutrition. Large herds of livestock are inappropriate to achieve these goals due to the relevant impact of meat supply chain on the environment, (e.g. land used to grow feed for animals is eight times more than that for human nutrition). The search for meat alternatives, especially for the intake of a critical nutrient as protein, is a consequent step. In the above context, this paper summarizes the health aspects of protein-rich food alternative to meat and carry out a literature review on the life-cycle environmental impacts of this alternative food”.

Reviewer 2: Introduction: In the last paragraph rewrite the objective of the review clearly.

Authors: Thank you for your comment. The last paragraph was modified as following: “In the above context, the goal of this paper is double:

-to describe the healthy characteristics of protein-rich food that can be used as alternative to meat (Section 2), by highlighting the advantages of this food as substituting of meat; and

-to carry out a literature review on the life cycle impacts of protein-rich food alternative to meat and the potential environmental benefits of substituting meat with other food giving the same amount of proteins (Section 3).

Reviewer 2: Introduction: Line no. 105-107 confusing statement.

Authors: According to your suggestion, the sentence was modified as following: “the International Agency for Research on Cancer (IARC) recently classified processed meat (meat transformed through salting, curing, fermentation, or smoking) as carcinogenic and red meat (beef, veal, pork, lamb, mutton, horse and goat) as probably carcinogenic [30]”.

Reviewer 2: Introduction: Line No. 131-134 what it mean?

Authors: Thank you for your comment. Authors decided to eliminate the sentence because it is out of the context of the paper.

Reviewer 2: Introduction: Need to present more data on specific meat alternative plant-based protein sources.

Authors: According to your suggestion, authors added the following sentence: “Soybean, peas, quinoa, rice, green beans, faba beans, lentils, lupine are among food that can be used as protein source to be used as alternative to meat”.  

Reviewer 2: Introduction: The LCA methods were well discussed and compared the published methods.

Authors: Thank you for your comment.

Reviewer 3 Report

The manuscript entitled “Life cycle environmental impacts and health effects of protein rich food alternative to meat” (Manuscript ID: sustainability-1560178) is an interesting review concerning the health aspects of protein-rich food alternative to meat and carry out a literature review on the life-cycle environmental impacts of this alternative food using several indicators such as GWP, land use, energy consumption and water impacts. Despite very significant variability in methodology and data concerning the life cycle environmental impacts extracted from 22 LCA case studies (FUs and system boundaries); the authors succeeded to carry out an interesting analysis and concluded that the plant-based high-protein food is the more environmental-friendly choice compared to meat.

I think that the quality of the manuscript can improve by considering a very important point in the first part (Health characteristics of animal vs. plant-based, protein-rich foods):

In this first part “Health characteristics of animal vs. plant-based, protein-rich foods” the authors focalized essentially on the positive effects of soy products, which contain phytoestrogens. In the literature, health benefits of soy products including a lowered risk of osteoporosis, heart disease, breast cancer, and menopausal symptoms, are frequently attributed to phytoestrogens but many are also considered endocrine disruptors, indicating that they have the potential to cause adverse health effects as well (The pros and cons of phytoestrogens. By: Heather B. Patisaul* and Wendy Jefferson. doi: 10.1016/j.yfrne.2010.03.003).

Minor concerns:

Line 258. Zhu and Ierland (2014): 2004 instead to 2014 (Xueqin Zhu & Ekko C. van Ierland (2004) Protein Chains and Environmental Pressures: A Comparison of Pork and Novel Protein Foods, Environmental Sciences, 1:3, 254-276, DOI: 10.1080/15693430412331291652).

Author Response

Response to the Reviewers

The authors would like to thank the Reviewers for their thorough analysis that will for sure help in improving the scientific quality of the paper. In the following pages, the comments of the Reviewers are reported and, for each comment, a reply from the authors is included in the text.

Reviewer 3: The manuscript entitled “Life cycle environmental impacts and health effects of protein rich food alternative to meat” (Manuscript ID: sustainability-1560178) is an interesting review concerning the health aspects of protein-rich food alternative to meat and carry out a literature review on the life-cycle environmental impacts of this alternative food using several indicators such as GWP, land use, energy consumption and water impacts. Despite very significant variability in methodology and data concerning the life cycle environmental impacts extracted from 22 LCA case studies (FUs and system boundaries); the authors succeeded to carry out an interesting analysis and concluded that the plant-based high-protein food is the more environmental-friendly choice compared to meat.

Authors: Thank you for the appreciation. Authors modified the paper according to your suggestions and comments.

Reviewer 3: I think that the quality of the manuscript can improve by considering a very important point in the first part (Health characteristics of animal vs. plant-based, protein-rich foods):

In this first part “Health characteristics of animal vs. plant-based, protein-rich foods” the authors focalized essentially on the positive effects of soy products, which contain phytoestrogens. In the literature, health benefits of soy products including a lowered risk of osteoporosis, heart disease, breast cancer, and menopausal symptoms, are frequently attributed to phytoestrogens but many are also considered endocrine disruptors, indicating that they have the potential to cause adverse health effects as well (The pros and cons of phytoestrogens. By: Heather B. Patisaul* and Wendy Jefferson. doi: 10.1016/j.yfrne.2010.03.003).

Authors: Thank you for the comment. The suggested point was discussed in the text: “In this context, it is noteworthy to mention that some phytoestrogens as from soy might have opposing health effects as they are classified endocrine disruptors [52]. However, recent data show that the impact on male reproductive hormones is low or absent [53] or even have tendentially beneficial effects on the risk of prostate cancer [54] and breast cancer mortality and cancer recurrence [55]”.

Reviewer 3: Minor concerns: Line 258. Zhu and Ierland (2014): 2004 instead to 2014 (Xueqin Zhu & Ekko C. van Ierland (2004) Protein Chains and Environmental Pressures: A Comparison of Pork and Novel Protein Foods, Environmental Sciences, 1:3, 254-276, DOI: 10.1080/15693430412331291652).

Authors: Thank you for highlighting this mistake, which was corrected both in the text and in the reference list.